

# Geographic genetic variation in the Coral Hawkfish, *Cirrhitichthys oxycephalus* (Cirrhitidae), in relation to biogeographic barriers across the Tropical Indo-Pacific

Rolando Quetzalcoatl Torres-García[1,2], Michelle R. Gaither[3],
D. Ross Robertson[4], Eloisa Torres-Hernández[5], Jennifer E. Caselle[6],
Jean-Dominique Durand[7], Arturo Angulo[8], Eduardo Espinoza-Herrera[9],
Francisco J. García-De León[10], Jonathan Valdiviezo-Rivera[11] and
Omar Domínguez-Domínguez[1,11]

[1] Laboratorio de Biología Acuática, Facultad de Biología, Universidad Michoacana de San Nicolás de Hidalgo, Morelia, Michoacán, Mexico
[2] Programa Institucional de Maestría en Ciencias Biológicas, Universidad Michoacana de San Nicolás de Hidalgo, Morelia, Michoacán, Mexico
[3] Department of Biology, Genomics and Bioinformatics Cluster, University of Central Florida, Orlando, Florida, United States
[4] Smithsonian Tropical Research Institute, Balboa, Panama, Panama
[5] Colección Nacional de Peces, Pabellón Nacional de la Biodiversidad, Departamento de Zoología, Instituto de Biología, Universidad Nacional Autónoma de México, Ciudad de México, Mexico
[6] Marine Science Institute, University of California, Santa Barbara, Santa Barbara, California, United States
[7] MARBEC, Université de Montpellier-IRD-CNRS, Montpellier Cedex, France
[8] Escuela de Biología, Museo de Zoología/Centro de Investigación en Biodiversidad y Ecología Tropical (CIBET), y Centro de Investigación en Ciencias del Mar y Limnología (CIMAR), Universidad de Costa Rica, San José, San José, Costa Rica
[9] Investigación Marina Aplicada, Parque Nacional Galápagos, Puerto Ayora, Isla Santa Cruz, Ecuador
[10] Laboratorio de Genética para la Conservación, Centro de Investigaciones Biológicas del Noroeste S. C., La Paz, Baja California Sur, Mexico
[11] Colección de Peces, Instituto Nacional de Biodiversidad, Quito, Pichincha, Ecuador

Corresponding author
Omar Domínguez-Domínguez,
omar.dominguez@umich.mx

## ABSTRACT

The Tropical Indo-Pacific (TIP) includes about two thirds of the world's tropical oceans and harbors an enormous number of marine species. The distributions of those species within the region is affected by habitat discontinuities and oceanographic features. As well as many smaller ones, the TIP contains seven large recognized biogeographic barriers that separate the Red Sea and Indian Ocean, the Indian from the Pacific Ocean, the central and eastern Pacific, the Hawaiian archipelago, the Marquesas and Easter Islands. We examined the genetic structuring of populations of *Cirrhitichthys oxycephalus*, a small cryptic species of reef fish, across its geographic range, which spans the longitudinal limits of the TIP. We assessed geographic variation in the mitochondrial *cytb* gene and the nuclear *RAG1* gene, using 166 samples collected in 46 localities from the western to eastern edges of the TIP. Sequences from *cytb* show three well-structured groups that are separated by large genetic distances (1.58–2.96%): two in the Tropical Eastern Pacific (TEP), one at Clipperton Atoll another occupying the rest of that region and the third that ranges across the remainder of the TIP, from the central Pacific to the Red Sea and South

Africa. These results indicate that the ~4,000 km wide Eastern Pacific Barrier between the central and eastern Pacific is an efficient barrier separating the two main groups. Further, the ~950 km of open ocean that isolates Clipperton Atoll from the rest of the TEP is also an effective barrier. Contrary to many other cases, various major and minor barriers from the Central Indo-Pacific to the Red Sea are not effective against dispersal by *C. oxycephalus*, although this species has not colonized the Hawiian islands and Easter Island. The nuclear gene partially supports the genetic structure evident in *cytb*, although all haplotypes are geographically mixed.

## INTRODUCTION

Contemporary and historical habitat discontinuities are fundamental drivers of geographic patterns of genetic structure and differentiation among natural populations. The wide range of dispersal strategies found among marine species is presumed to influence micro- and macro-evolutionary process (*Kinlan & Gaines, 2003*; *Bullock, Shea & Skarpaas, 2006*; *Riginos et al., 2014*). In addition to biological aspects, factors such as oceanographic conditions and/or habitat availability, may limit or enhance dispersal (*Lessios, Kessing & Pearse, 2001*; *Luiz et al., 2012*; *Sandoval-Huerta et al., 2019*; *Palmerín-Serrano et al., 2021*; *Torres-Hernández et al., 2022a*, *2022b*). Over time, the geographic barriers that limit the movement of individuals between populations can lead to the evolution of discrete phylogenetic clades (*Taylor & Hellberg, 2006*).

The Tropical Indo-Pacific (TIP), the vast, ~28,000 km wide, region that stretches from the Rea Sea to west coast of the American continent, is home to an enormous diversity of marine organisms. Numerous studies have demonstrated that many marine species in the TIP show patterns of genetic structure that reflect limitations on dispersal by historical geographical or oceanographic barriers, suggesting that the fish diversity is underestimated due to the presence of species complexes in widely distributed groups (*Lessios et al., 1999*; *Lessios, Kessing & Pearse, 2001*; *Sandoval-Castillo et al., 2004*; *Lessios & Robertson, 2006*; *Zemlak et al., 2009*).

Within the TIP seven major biogeographical barriers have shaped the distributions of tropical marine species. The first is the Red Sea Barrier, represented by the shallow Strait of Bab al Mandab which reduces water exchange between the Red Sea and the Indian Ocean during glacial maxima (*Rohling et al., 1998*; *Siddall et al., 2003*; *Bailey, 2009*; *DiBattista et al., 2016*). Second, the Indo-Pacific Barrier, a recognized partition in the Indo-Malayan Archipelago that separates the Indian and Pacific Oceans (*Briggs, 1974*; *Gaither et al., 2011b*; *Gaither & Rocha, 2013*), reflects effects of repeated episodes of lowered sea level during the Plio-Pleistocene glaciations, when the emergence of the Sunda Shelf separated those two oceans (*Voris, 2000*; *Reid et al., 2006*). The third is the mid-Indian Ocean oceanic barrier, the large deep-water area west of the Lakshadweep-Maldives-Chagos

archipelagoes (*Borsa et al., 2016*). The fourth is the Marquesas Barrier, formed by the geographical isolation and unusual oceanographic conditions (*Randall, 2001*; *Gaither et al., 2010*). The fifth, the Hawaiian Barrier, attributed to the island isolation and ocean currents (*Gaither et al., 2011a*). The sixth is the deep water isolating the Easter Island Province (*sensu Briggs & Bowen, 2012*) from French Polynesia. Finally, the Eastern Pacific Barrier (EPB), the world's widest (4,000 km) deep-water barrier, separates the Tropical Eastern Pacific (TEP) from the remainder of the TIP (*Robertson, Grove & McCosker, 2004*). *Crandall et al. (2019)* examined various schemes proposed for biogeographic and provincial subdivisioning of the TIP in relation to the existence of those large scale, and other, smaller-scale barriers in a variety of reef organisms. The region known as the Tropical Indo-Central Pacific (TICP) includes the Red Sea and Western Indian Ocean provinces (WIOP) in the Indian Ocean, the Indo-Polynesian, Hawaiian, Marquesan (IP-H-M) and the Easter Island Provinces in the area stretching from the west coast of Australasia to the islands bordering the western edge of the EPB. This represents the scheme of *Briggs & Bowen (2012)*, with the addition of the Red Sea Province (RSP). The TEP includes the Cortez and Panamic Provinces on the mainland and the Oceanic Island Province (*Robertson & Cramer, 2009*; *Briggs & Bowen, 2012*). Environmental heterogeneity within the TEP has allowed for the development of unique local evolutionary processes, which are reflected in various patterns of geographic genetic differentiation within widely distributed species in the region (*Hastings, 2000*; *Mora & Robertson, 2005*; *Sandoval-Huerta et al., 2019*; *Palmerín-Serrano et al., 2021*; *Torres-Hernández et al., 2022a*, *2022b*).

Few species span the entire TIP. One such species is the Coral Hawkfish, *Cirrhitichthys oxycephalus* (Bleeker, 1855), an ideal subject for examining the effects of these different biogeographic barriers because its geographic range extends across the entire TIP (*Allen & Erdmann, 2012*) except the Hawaiian (NE central Pacific) and Pitcairn-Easter Island (SE Central Pacific) Provinces where it is absent (Fig. S1) (*Randall, 1999*, *2007*; *Randall, Cea & Meléndez, 2005*). *Cirrhitichthys oxycephalus* is a cryptobenthic reef fish typically found at depths of less than 40 m (*Lieske & Myers, 1994*). Like other hawkfishes, *C. oxycephalus* is sexually dimorphic in body size (males are larger) (*Donaldson, 1988*) and likely is a protogynous hermaphrodite (*Sadovy & Donaldson, 1995*). Populations are organized in social groups or harems in which there is a dominant male and one or more females of different sizes. Each social group defends a territory with individuals resting on a variety of substrata, especially within branching corals of the genus *Pocillopora*, which are common shallow-water corals throughout the TIP range of the Coral Hawkfish. *Cirrhitichthys* species spawn pelagic eggs, and the duration of the pelagic larval stage of *C. oxycephalus* is about 36 to 51 days (*Brothers & Thresher, 1985*; *Robertson, Grove & McCosker, 2004*). Although the species is considered a non-obligated, coral-dwelling species, this may vary with geographic location (*Donaldson & Myers, 1988*; *Donaldson, 1989*; *Robertson & Allen, 1996*; *Palacios & Zapata, 2014*; our observations in the TEP). For example, at Cabo Pulmo reef in the TEP the species is considered to be live-coral dependent (*Alvarez-Filip & Reyes-Bonilla, 2006*), whereas in Chagos in the IP-H-M, *C. oxycephalus* is found on pavement habitats rather than live-coral habitat (*Coker et al., 2015*).

The distribution of *C. oxycephalus* across almost the entire TIP (except the Hawaiian and Easter Island Provinces (*sensu Briggs & Bowen, 2012*; see *Randall, 1999*, *2007*; *Randall, Cea & Meléndez, 2005*)), along with its life history traits raise questions about the effects of known major biogeographic barriers on the geographic genetic structuring of its populations and their taxonomical status. Accordingly, using mitochondrial and nuclear markers, we analyzed the genetic structure of *C. oxycephalus* throughout all major parts of its TIP range to (a) determine the extent and patterns of genetic connectivity or fragmentation across its enormous geographic distribution, and (b) understand the influence of large- and small-scale habitat discontinuities and oceanographic processes on the distribution of genealogical lineages. We expected to find a phylogeographic pattern with various isolated populations due to this species' association with coral reefs, the depth range at which its lives, and its pelagic larval duration (PLD), and because geographic barriers represented by expanses of deep water in various parts of the TIP are known to have promoted genetic differentiation in many reef organisms, including many fishes. Futhermore, *Lessios & Robertson (2006)* examined trans-EPB genetic relationships of populations of *C. oxycephalus*, using two mitochondrial genes (different to that used here) and concluded that the Coral Hawkfish originated in the TEP and spread westward to the TICP. A more comprehensive dataset involving data from another gene and populations scattered throughout its entire geographic range should provide information relevant to that conclusion.

## MATERIALS AND METHODS

### Sample collection

We collected 166 individuals of Coral Hawkfish from 45 sites at 20 locations found in all biogeographic regions and provinces from the TIP (Fig. 1 and Table S1). Specimens were collected on SCUBA using, in different locations and at different times, various combinations of rotenone ichthyocide (*Robertson & Smith-Vaniz, 2008*), clove oil anesthetic (*Piñeros et al., 2019*) and either aquarium hand nets, a suction tool or pole-spears. We preserved tissue samples in either DMSO buffer or 95–96% ethanol and stored them at either −20 °C or −76 °C. Specimens were deposited in the fish collections at the Universidad Michoacana de San Nicolás de Hidalgo, Mexico, University of Central Florida, USA, The National Museum of Natural History of the Smithsonian Institution, USA, the King Abdullah University of Science and Technology, Saudi Arabia, University of California Santa Barbara, USA, and Ho Chi Minh City University of Science, Vietnam (Table S1). Organism collection was supported and allowed by the following institutions and permits by Mexican Ministry of Environment and Natural Resources under collection permits numbers (PPF/DGOPA-035/15; and F00.DRPBCPN.DIR.RBAR-100/2015-CONANP), and for El Salvador (MARN-AIMA-004-2013), Panama (SC/A-17-19), Costa Rica (007-2013-SINAC and R-056-2105-OT-CONAGEBIO), and Ecuador (013/2012 PNG; N°21-2017-EXP-CM-2016-DNB/MA and MAAE-DBI-CM-2021-0152).

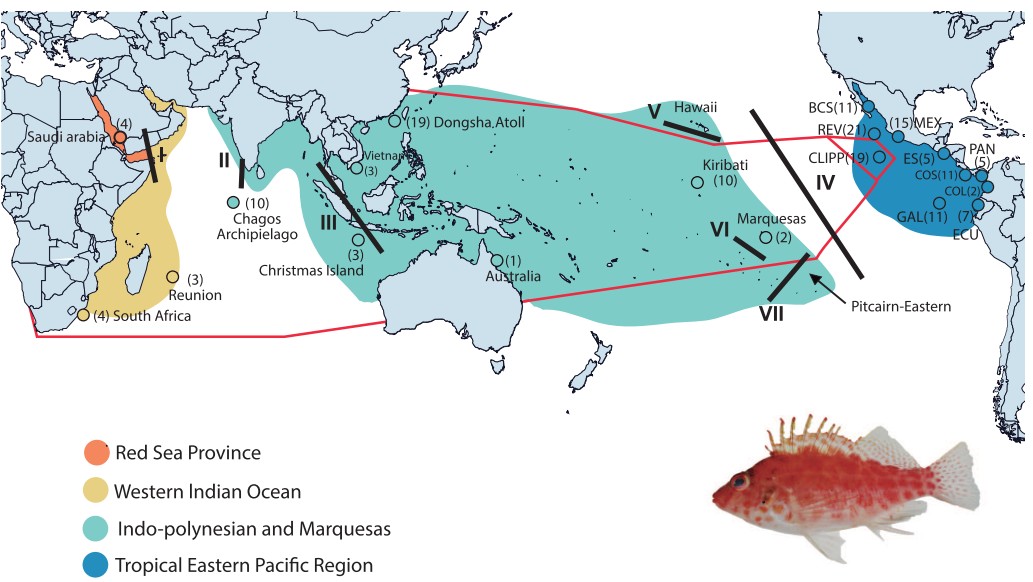

**Figure 1 Collection areas along the Tropical Indo-Pacific (TIP).** Biogeographical provinces and regions (*sensu Briggs & Bowen, 2012*; but with the addition of the Red Sea Province) are shown in colors. Numbers in parentheses are samples sizes for each collection site. The red lines represent the hypothetic barriers of mitochondrial gene shown by the K = 3 with SAMOVA. Lines in black represent the positions of the biogeographic barriers: I = Red Sea Barrier, II = mid-Indian Ocean oceanic Barrier; III = Indo Pacific Barrier, IV = Eastern Pacific Barrier, V = Hawaiian Barrier, VI = Marquesas Barrier, VII = Eastern Island Barrier. BCS = Baja California Sur; REV = Revillagigedo Islands; MEX = central Mexico; CLIPP = Clipperton; ES = El Salvador; PAN = Panama; COS = Costa Rica; COL = Colombia; GAL= Galapagos; ECU = Ecuador (for detailed locality information see Table S1). *C. oxycephalus* is present throughout the map areas except for the Hawaiian and Pitcairn-Easter Island Province. Map was done in Qgis, base map layer is from http://datacatalog.worldbank.org/search/dataset/0038272/World-Bank-Official-Boundaries. Modified to World Country Polygons-Very High-Definition (zip file wb_countries_admin0_10m) CC-BY 4.0. Photo credit: Francisco Martínez-Servín.

## DNA extraction and sequencing

We extracted total genomic DNA from tissue samples using the phenol-chloroform protocol of *Sambrook, Fritsch & Maniatis (1989)*. We PCR amplified a 652 base pair (bp) fragment of the mitochondrial cytochrome b (*cytb*) gene, using the primers GluDG (forward) and H16460 (reverse) (*Perdices & Doadrio, 2001*). Because the samples from Clipperton failed to amplify using these universal primers, we designed specific primers 1F31 ACGGCTGACTAATCCGTA (forward) and 1R61 AATTAGGGATGCGA CTTGTCC (reverse). Taking into account the degree of variation found in the *cytb*, we also amplified a 132 bp fragment of the recombination-activating nuclear gene 1 (*RAG1*) in a subsample of 45 samples using the primers *RAG* 1F (forward) and *RAG* 9R (reverse) (*Quenouille, Bermingham & Planes, 2004*). We performed all PCRs in a total volume of 12.5–16.5 μL with 1 μL (50–100 ng) DNA template, 0.25 μl of 10 μM for each primer, 0.2–0.5 μl of 50 mM MgCl2, 0.25–0.9 μl of 10 mM dNTPs, 0.0625–0.088 μL of 5 U/μL Taq polymerase (Invitrogen, Waltham, MA, USA), 1.25–1.5 μL of 10X buffer and deionized sterile water to reach the final volume. Those PCRs utilized the following thermocycling

conditions: initial denaturation at 94 °C (3 min) and a final extension at 72 °C (10 min), with an intervening 35 cycles of 30 s at 94 °C, 45 s at the annealing temperature (51 °C for *cytb*; 59 °C for *RAG1*), and 1 min at 72 °C. We visualized the PCR products on 1.5% agarose gels and sent amplicons to MACROGEN Inc. (Seoul, South Korea) for sequencing.

We edited and aligned all the sequences using chromatograms with the software MEGA v7.0.2 (*Kumar, Stecher & Tamura, 2016*). We conducted the recombination test for the nuclear genes with the software Split Tree v4 (*Huson & Bryant, 2006*) for the *RAG1* gene. We deposited *cytb* unique sequences in GenBank (Table S1) and sequences of *RAG1* are available at https://doi.org/10.5281/zenodo.11323581.

## Data analyses
### Phylogenetic analyses and haplotype networks
For our phylogenetic analyses, we determined the best fit evolutionary model of substitution for the mitochondrial (*cytb*) and nuclear (*RAG1*) genes using the Akaike Information Criteria (AIC) and the optimal partition setting using PartitionFinder v.1.1.0 (*Lanfear et al., 2012*). We conducted phylogenetic reconstructions using a Maximum Likelihood (ML) approach and Bayesian Inference (BI) for each gene separately. ML analyses were conducted using RaxMLGUI v8 (*Stamatakis, 2014*) and the Generalized Time Reversible substitution model (GTR) with gamma distribution, invariable sites, and 10,000 bootstrap replicates. We generated the Bayesian Inference (BI) reconstruction using MrBayes v3.2.6 (*Ronquist et al., 2012*) with the substitution model GTR + Gamma + I, and ran it for 50 million generations, with two independent runs implementing four Markov Chain Monte Carlo (MCMC) processes and sampling every 1,000 generations. We evaluated the chains convergence using effective sample sizes (ESS) >200 for all parameters in Tracer v1.6 (*Rambaut et al., 2015*) and discarding the initial 10% of the generations as burn-in to construct the consensus tree ($\sigma = 0.0002$). We included *Cirrhitichthys aprinus* (Cuvier, 1829) as the outgroup. MrBayes runs were preformed using the CIPRES portal (*Miller, Pfeiffer & Schwartz, 2010*).

To examine the relationships between sequences, we constructed a statistical parsimony network for each gene using the Median-Joining algorithm in the software PopArt v.1.7 (http://popart.otago.ac.nz).

### Genetic diversity, genetic distances, and genetic structure
To determine levels of genetic variability partitioned within and among populations an analysis of molecular variance (AMOVA) was performed in ARLEQUIN (*Excoffier, Smouse & Quattro, 1992*; *Excoffier & Lischer, 2010*) for the two loci separately, with the application of a Bonferroni correction (*Rice, 1989*) to correct for multiple comparisons. The analysis was carried out (1) with all samples without grouping, (2) partitioned into the three groups found in the haplotype network, (3) according to the modified regionalization of *Briggs & Bowen (2012)* (Red Sea Province = RSP/Western Indian Ocean Province = WIOP/Indo-Polynesian and Marquesas Island Provinces = IP-M/TEP Region) and (4) according to sampled location/regions.

We also ran a spatial analysis of molecular variance (SAMOVA 1.0) (*Dupanloup, Schneider & Excoffier, 2002*) for the *cytb* dataset for values of K = 2 to 10 (*Eble et al., 2011*). Due to the lack of spatial resolution in the *RAG1* haplotype network, we did not run the SAMOVA for this gene. We calculated uncorrected genetic *p*-distances (*pd*) and $\Phi$ST using MEGA v7.0 (*Kumar, Stecher & Tamura, 2016*) for pairwise comparison, according with sampled location/regions, *Briggs & Bowen (2012)* regionalization, and the tree genetic groups found in haplotype network and supported by the genetic structure analyses. An isolation by distance (IBD) analysis was performed using the Mantel test on the R platform v4.1.2 using the packages vegan (*Oksanen et al., 2018*) and fossil (*Goslee & Urban, 2007*), to verify the correlation between the genetic distance matrixes (estimated by pairwise $\Phi$ST) and geographical distances. We ran the IBD tests using (1) all the samples and (2) samples separated into two groups: the TICP and TEP samples. We estimated the number of haplotypes (*hn*), segregating sites (*SS*), haplotype diversity (*h*), nucleotide diversity ($\pi$) in relation to the sampled location/regions, the modified *Briggs & Bowen (2012)* regionalization and for the three groups resolved in the haplotype networks and the AMOVA and SAMOVA analyses using the software DnaSP v.5.0 (*Librado & Rozas, 2009*).

## RESULTS

We resolved 166 *cytb* sequences (652 bp) that generated 108 haplotypes and shows 163 polymorphic segregating sites, out of which 89 were variable, 68 were parsimoniously informative and 84 were singleton variable sites. For *RAG1* (132 bp) we obtained 90 sequences (including the two alleles) and 17 genotypes. For that gene we found 12 polymorphic segregation sites, six of which were parsimoniously informative, and six singleton variables sites. We found no evidence of recombination in this nuclear gene (*RAG1* = 0.052 *p* > 0.05).

### Phylogenetic analyses and haplotype networks

The phylogenetic analysis for *cytb* shows a basal polytomy that includes all 59 samples from the various provinces within the TICP and another group composed of all 107 TEP samples. In the TEP group there are two sub-clades, one represented by all but one of 19 samples from Clipperton, plus four of 21 samples from Revillagigedo Archipelago and two of 25 from Mainland Mexico, the CL-TEP sub-clade. The second sub-clade (TEP), which had low support, includes the remaining 82 samples from the (23 samples from Mexico mainland, 17 from the Revillagigedo Archipelago, five from El Salvador, one from Costa Rica mainland, nine from Cocos Island, six from Panama, two from Colombia mainland, seven from Ecuador mainland, 12 from the Galapagos Archipelago) and one from Clipperton Atoll (Fig. 2). The phylogenetic analyses for the *RAG1* gene show a basal polytomy without resolution (see Fig. S2).

The *cytb* haplotype network resolved three haplogroups. One haplogroup (TICP, Fig. 3) includes all the samples from the TICP and includes a mixture of haplotypes from the different regions and provinces, with 1–10 mutation steps separating different haplotypes from their closest neighbors. The TICP haplogroup is separated by 10 mutational steps from haplotypes in the main TEP haplogroup (Fig. 3), which included 44 of 46 samples

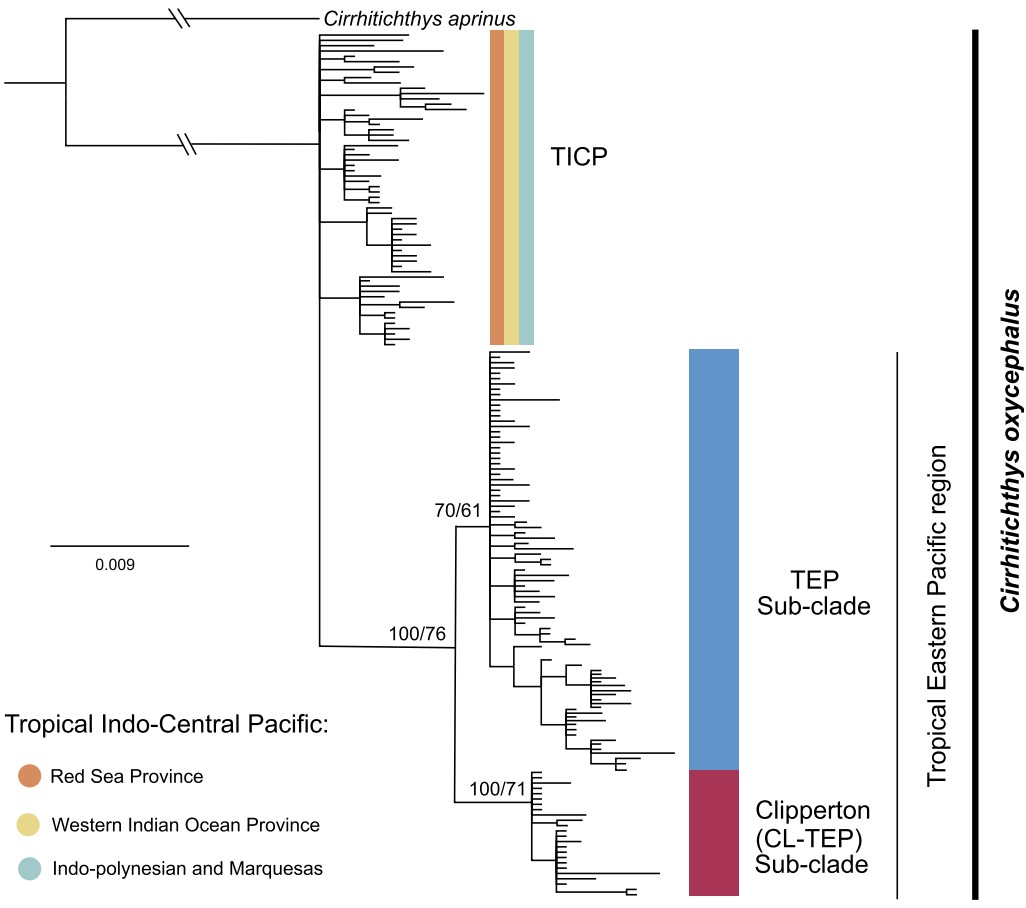

**Figure 2** **Phylogenetic tree based on the** *cytb* **gene and ML and BI.** Bootstrap/posterior probability supports are shown in the nodes. Colors represent the different haplogroups (Hg) found in the haplotype network (Fig. 3). TICP: Tropical Indo-Central Pacific; TEP: Tropical Eastern Pacific region except for CL-TEP: Clipperton-TEP.                                

from the mainland TEP, one of 19 samples from Clipperton Atoll, 17 of 21 samples from the Revillagigedo Archipelago, and all nine samples from Cocos Island and 12 from the Galapagos Archipelago. The TEP haplogroup in turn was separated by six mutation steps from the Clipperton haplogroup (CL-TEP). In contrast to *cytb* haplotype network, *RAG1* resolved no geographic structuring and the two most common alleles in this dataset were represented in all regions and provinces (Fig. 4).

## Genetic structure, distances, and genetic diversity

In general, AMOVA results for *cytb* found the highest genetic variation among groups, with all comparisons being statistically significant (Table 1). The arrangement of three groups (TICP/TEP/CL-TEP) showed the highest and significant value of $\Phi CT$ (64.39%) (Table 1). Results for *RAG1* should be taken with caution due to the lack of samples in the Red Sea Province (representing the only one of four total regions without *RAG1* samples), which we were unable to amplify. The highest percentage and statistically significant values for *RAG1* were for the within populations (89.19%) comparison in the three-group arrangement (Western Indian Ocean Province/Indo-Polynesian and Marquesas Island
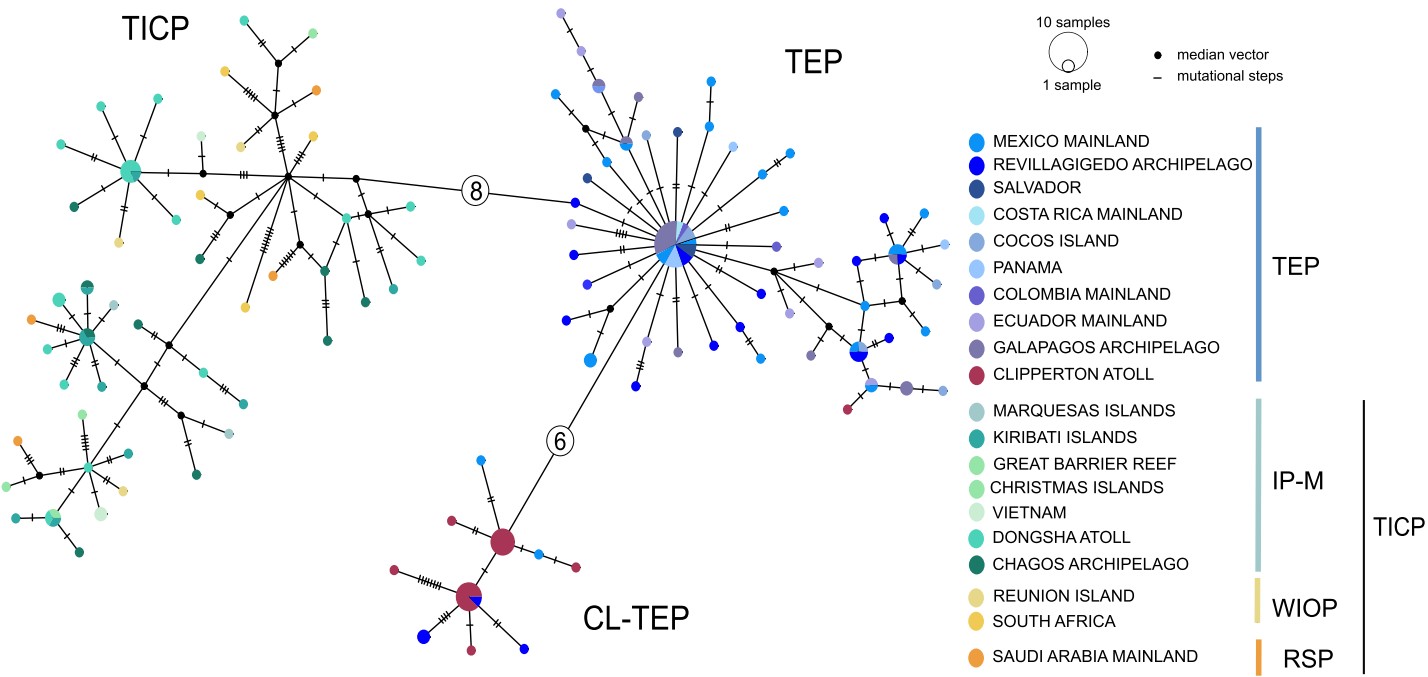

**Figure 3** **Median-joining networks for *cytb*.** Sizes of the circles indicate the frequency of the haplotype. Colors in circles correspond to the areas where the samples were collected. Each dash or numbers in the white circle represent the number of mutation steps. TICP: Tropical Indo-Central Pacific region (*sensu Briggs & Bowen, 2012*, with the addition of the Red Sea Province), which includes the IP-M: the East Indian Ocean plus Indo-Polynesian Province plus the Marquesan, the WIOP: Western Indian Ocean Province; the RS: Red Sea Province; and the TEP: Tropical Eastern Pacific region, which includes CL-TEP: Clipperton-TEP.

Province/TEP Region; Table 1). For *cytb*, the K = 3 arrangement maximized the differences between groups in the SAMOVA analysis with the highest and significant value of $\Phi$CT (59.36%), segregating the populations into three groups: TICP, TEP and CL-TEP (Fig. 1 and Table 2). We obtained high and significant levels of genetic differentiation among those three groups. The highest and most significant $\Phi$ST value for *cytb* was obtained between TICP and CL-TEP groups ($\Phi$ST = 0.637), and the lowest between TEP and CL-TEP ($\Phi$ST = 0.493). The largest genetic distance (*pd*) detected among groups also was between the TICP and CL-TEP (2.96%), and the lowest between the TEP and CL-TEP (1.58%). For *RAG1* the highest *pd* genetic distance was between TEP and TIP (1.3%) and the lowest between CL-TEP and TIP (0.9%), while the highest and significant $\Phi$ST value was between TEP and CL-TEP (0.114) and the lowest between TIP and CL-TEP (0.012) (Table 3). For location/regions comparisons involving *cytb* the highest and significant $\Phi$ST was obtained between Galapagos and Vietnam (0.818), and the lowest between RSP and WIOP (−0.02), while the highest *pd* distances was obtained between Mexican and Arabian populations (3.228%) and the lowest between the Cortez and Panamic Provinces of the TEP (0.001). For *RAG1* the highest *pd* was between Mexican and Christmas populations (1.799%) and lowest between the Galapagos and Clipperton populations (0.379), whereas the highest and significant $\Phi$ST (0.298) was between Mexican and Clipperton populations and the lowest between Dongsha and Kiribati populations (0.001) (for more details see Table S2). For the modified *Briggs & Bowen (2012)* 4-area scheme the highest *pd* was

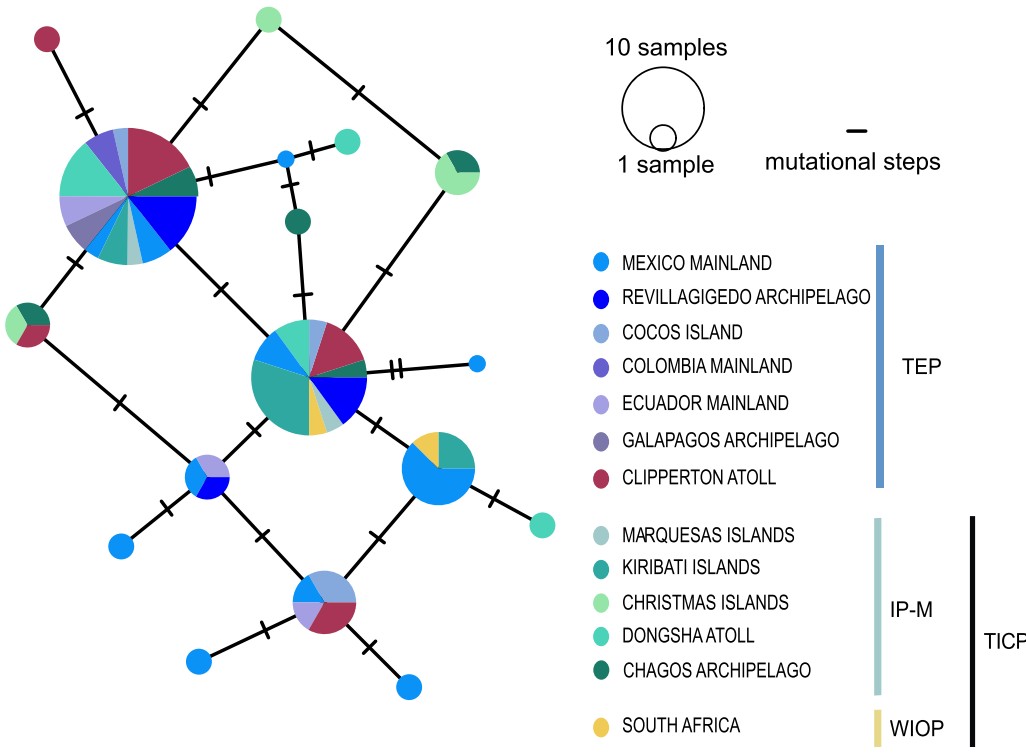

**Figure 4 Median-joining networks for *RAG1* gene.** Size of the circles indicate the frequency of the haplotype. Colors in circles correspond to the areas where the samples were collected. Each dash represents the mutation steps. TEP, Tropical Eastern Pacific region; IP-M, East Indian Ocean, Indo-Polynesian and Marquesas Province; WIOP, Western Indian Ocean Province.

between the RSP and TEP (2.94%), and the lowest between RSP and IP-M (1.47%), with the highest and significant $\Phi_{ST}$ obtained between RSP and TEP (0.617) and the lowest between RSP and WIOP (−0.029) (Table S3).

The Mantel test showed a significant correlation between genetic- and geographic distances across the entire TIP dataset (r = 0.49, $p < 0.05$) but not within either the TICP (r = 0.11, $p > 0.05$) or the TEP (r = 0.05, $p > 0.05$).

Overall, nucleotide diversity ($\pi$) for *cytb* was 0.017 and haplotype diversity (*h*) was 0.976, with values of 0.011 and 0.803, respectively, for *RAG1*. For the three groups resolved in the haplotype networks CL-TEP shown the highest values for haplotype diversity (*cytb h* = 0.989 +/− 0.006 and *RAG1 h* = 0.827 +/− 0.046) and TEP the lowest values for nucleotide diversity (*cytb $\pi$* = 0.754 +/− 0.071 and *RAG1 $\pi$* = 0.703 +/− 0.101) (Table 4). For genetic diversity according to the *Briggs & Bowen (2012)* plus Red Sea regionalization, the highest haplotype diversity values were from the Red Sea Province (*cytb h* = 1 +/− 0.177) and the Western Indian Ocean Province (*RAG1 h* = 1 +/− 0.500). The lowest value was for Australia (*cytb h* = 0) and Arabia, Vietnam, Australia and the Galapagos (*RAG1 h* = 0) (Table 5). For genetic diversity according with location/regions, the highest values were from the Mexican Province (*cytb h* = 0.989 +/− 0.031) and from South Africa and the

**Table 1 Molecular variance.** AMOVA results for the mitochondrial *cytb* gene and nuclear *RAG1* gene.

| | N | Groups | ΦST | ΦCT | ΦSC | Within populations | Among groups | Among populations within groups |
|---|---|---|---|---|---|---|---|---|
| *Cytb* | 1 | Oveall | 0.564*** | 0 | 0 | 43.58 | 0 | 56.42 |
| | 3 | Red Sea Province + Western Indian Ocean Province + Indo-Polynesian and Marquesas/TEP/CL-TEP | 0.662*** | 0.644*** | 0.05 | 33.82 | 64.39 | 1.78 |
| | 4 | Red Sea Province/Western Indian Ocean Province/Indo-Polynesian, and Marquesas /TEP Region | 0.642*** | 0.553*** | 0.198*** | 35.79 | 55.36 | 8.85 |
| | 16 | Saudi Arabia/SouthAfrica + Reunion/Christmas/Vietnam/ Dongsha/Australia/Kiribati/Chagos/Marquesas/Revillagigedo/ Cortez/Mexican/Panamic/Cocos/Clipperton/Galapagos | 0.504*** | 0.534*** | −0.063 | 49.54 | 53.42 | −2.96 |
| *RAG1* | 1 | Oveall | 0.121 | 0 | 0 | 87.93 | 0 | 12.07 |
| | 3 | Western Indian Ocean Province + Indo-Polynesian and Marquesas/TEP/CL-TEP | 0.142* | 0.056 | 0.090 | 85.77 | 5.65 | 8.58 |
| | 3 | Western Indian Ocean Province/Indo-Polynesian and Marquesas/TEP Region | 0.108** | 0.039 | 0.072* | 89.19 | 3.91 | 6.90 |
| | 13 | SouthAfrica/Christmas/Dongsha/Australia/Kiribati/Chagos/ Marquesas/Revillagigedo/Cortez/Mexican/Panamic/Cocos/ Clipperton/Galapagos | 0.123** | 0.05 | 0.077 | 87.7 | 4.94 | 7.36 |

Note:
One asterisk = $p < 0.05$, two asterisks = $p < 0.01$ and three asterisks = $p < 0.001$. N, number of groups; TEP, Tropical Eastern Pacific (minus Clipperton); CL-TEP, Clipperton of the Tropical Eastern Pacific.

**Table 2 Spatial analysis of molecular variance.** SAMOVA results for the mitochondrial *cytb* gene.

| K | Groups | ΦST | ΦCT | ΦSC | Within populations | Among groups | Among populations within groups |
|---|---|---|---|---|---|---|---|
| 2 | TEP, CL-TEP/TICP | 0.643*** | 0.565*** | 0.178*** | 35.7 | 56.57 | 7.73 |
| 3 | TEP/CL-TEP/TICP | 0.611*** | 0.594*** | 0.043*** | 38.88 | 59.36 | 1.77 |
| 4 | TEP/CL-TEP/TIWP/KWA | 0.606*** | 0.593*** | 0.031*** | 39.39 | 59.33 | 1.28 |
| 5 | TEP/CL-TEP/TIWP/KWA/AUS | 0.601*** | 0.592*** | 0.021*** | 39.88 | 59.24 | 0.87 |
| 6 | TEP/CL-TEP/TIWP/KWA/AUS/ MAR | 0.591*** | 0.591**** | 0.018 | 40.13 | 59.14 | 0.74 |

Note:
One asterisk = $p < 0.05$, two asterisks = $p < 0.01$ and three asterisks = $p < 0.001$. TICP, Tropical Indo-Central Pacífic; TIWP, Tropical Indo-West Pacific; TEP, Tropical Eastern Pacific (minus Clipperton); CL-TEP, Clipperton of the Tropical Eastern Pacific; KWA, Kwa Zulu-Natal; AUS, Australia; MAR, Marquesas-Islands.

**Table 3 Genetics values.** Above diagonal pairwise population comparison (ΦST) and below diagonal mean genetic distances (*pd*) in percentage for *cytb* and *RAG1* genes.

| *cytb* | TICP | TEP | CL-TEP |
|---|---|---|---|
| TICP | 0 | 0.601* | 0.637* |
| TEP | 2.7 | 0 | 0.493* |
| CL-TEP | 2.96 | 1.58 | 0 |
| *RAG1* | TICP | TEP | CL-TEP |
| TICP | 0 | 0.094* | 0.012 |
| TEP | 1.3 | 0 | 0.114* |
| CL-TEP | 0.9 | 1.1 | 0 |

Note:
*Significant values ($p < 0.05$). TICP, Tropical Indo-Central Pacific; TEP, Tropical Eastern Pacific (minus Clipperton); CL-TEP, Clipperton of the Tropical Eastern Pacific.

**Table 4 Diversity indices.** Values for the mitochondrial *cytb* and nuclear *RAG1* genes according to the regionalization presented by *Briggs & Bowen (2012)*, with the addition of a Red Sea Province.

| | N | hn | SS | h | π |
|---|---|---|---|---|---|
| **cytb** | | | | | |
| Red Sea Province | 4 | 4 | 24 | 1 +/− 0.177 | 0.019 +/− 0.004 |
| Western Indian Ocean Province | 7 | 7 | 36 | 1 +/− 0.076 | 0.018 +/− 0.00223 |
| Indo-Polynesian and Marquesas | 48 | 38 | 67 | 0.987 +/− 0.008 | 0.011 +/− 0.001 |
| TEP Region | 107 | 60 | 97 | 0.95 +/− 0.014 | 0.010 +/− 0.001 |
| **RAG1** | | | | | |
| Red Sea Province | 0 | 0 | 0 | 0 | 0 |
| Western Indian Ocean Province | 2 | 2 | 2 | 1 +/− 0.500 | 0.015 +/− 0.008 |
| Indo-Polynesian and Marquesas | 30 | 11 | 9 | 0.825 +/− 0.048 | 0.011 +/− 0.002 |
| TEP Region | 58 | 10 | 7 | 0.773 +/− 0.041 | 0.011 +/− 0.001 |

**Note:**

N, number of individuals per haplogroup; hn, number of haplotypes; SS, segregating sites; h, haplotype diversity; π, nucleotide diversity.

**Table 5 Diversity indices.** Values for the mitochondrial *cytb* and nuclear *RAG1* genes in relation to the three groups resolved in the haplotype networks.

| | N | hn | SS | h | π |
|---|---|---|---|---|---|
| **cytb** | | | | | |
| TICP | 59 | 49 | 93 | 0.989 +/− 0.006 | 0.012 +/− 0.006 |
| TEP | 88 | 53 | 91 | 0.932 +/− 0.021 | 0.008 +/− 0.004 |
| CL-TEP | 19 | 7 | 25 | 0.754 +/− 0.071 | 0.004 +/− 0.002 |
| Overall | 166 | 108 | 163 | 0.976 +/− 0.006 | 0.017 +/− 0.008 |
| **RAG1** | | | | | |
| TICP | 32 | 11 | 9 | 0.827 +/− 0.046 | 0.011 +/− 0.001 |
| TEP | 44 | 9 | 6 | 0.785 +/− 0.042 | 0.012 +/− 0.001 |
| CL-TEP | 14 | 5 | 3 | 0.703 +/− 0.101 | 0.007 +/− 0.001 |
| Overall | 90 | 17 | 12 | 0.803 +/− 0.030 | 0.011 +/− 0.001 |

**Note:**

N, number of individuals per haplogroup; hn, number of haplotypes; SS, segregating sites; h, haplotype diversity; π, nucleotide diversity. TICP, Tropical Indo-Central Pacific; TEP, Rest of the Tropical Eastern Pacific; CL-TEP, Clipperton of the Tropical Eastern Pacific.

Marquesas (*RAG1 h* = 1 +/− 0.500). The lowest values were from Australia (*cytb h* = 0) and Arabia, Vietnam, Australia, and the Galapagos (*RAG1 h* = 0) (Table S4).

## DISCUSSION

The phylogenetic analyses, haplotype network, AMOVA, SAMOVA, genetic distances and ΦST analyses presented here for *cytb* of the Coral Hawkfish reveals the existence of two divergent and geographically structured clades, one distributed throughout in the TICP and other in the TEP. Within the TEP, two sub-clades were identified, one mainly restricted to Clipperton atoll and the other distributed primarily across the rest of the TEP (Figs. 2 and 3, Tables 1–3). For *RAG1* this differentiation was partially supported by the ΦST results, even though alleles were not distributed in a geographically restricted manner,

with the two most frequent alleles found in all biogeographic regions and provinces (Fig. 4 and Table 3).The discordance between the mitochondrial and nuclear genomes is a common phenomenon that occurs in many cases among animals, with a higher frequency in mammals and fishes than in other taxa (*Toews & Brelsford, 2012*). This may be due to differences in mutation rates between these two genes. For example, in 12 species of sharks from a range of genera the mutation rate for *RAG1* is substantially lower ($2.5 \times 10^{-10}$ yr$^{-1}$) than that of the mitochondrial *cytb* gene ($7.0 \times 10^{-10}$ yr$^{-1}$) (*Martin, 1999*). A similar pattern has been observed in various bony-fish groups (*Šlechtová, Bohlen & Perdices, 2008*; *Reece et al., 2011*; *Palmerín-Serrano et al., 2021*; *Pérez-Rodríguez et al., 2023*). Incomplete lineage sorting driven by the higher effective population size of *n*DNA could also have affected our results. Genetic drift, which promotes divergence between isolated populations, occurs four times slower in *n*DNA than *mt*DNA (*Hare, 2001*; *Larmuseau et al., 2010*) due to the lower effective population size of the mitochondrial genome. Another possibility is the relatively low number of base pairs utilized in the *RAG1* gene, coupled with the smaller sample size compared to the mitochondrial gene potentially could have introduced bias into our final results.

When we consider all 45 populations of Coral Hawfish we found a significant correlation between geographic and genetic distances. However, this dataset included samples on both sides of ~4,000 km wide EPB, which is an effective biogeographic barrier for many marine species. To test if our finding of IBD was driven by this barrier, we also assessed the situation within each of the two main clades, TICP and TEP, and found no indication of IBD in either dataset. According with these results, we consider that, except for effects of the EPB, geographic distance is not a major factor affecting genetic structure in *C. oxycephalus*.

## Two evolutionary lineages: an effect of the Eastern Pacific Barrier

The results presented here, based upon *mt*DNA, indicate that *C. oxycephalus* populations from two separate evolutionary units (lineages) (Fig. 3 and Tables 1–3), one distributed across the Tropical Indo-Central Pacific, and the other restricted to the Tropical East Pacific, separated by ten mutation steps, between 2.7 to 2.9 *pd* and with significant ΦST results. These results support a previous study by *Lessios & Robertson (2006)* using the mitochondrial ATPase six and eight genes, which indicated that the two regional clades are relics of long past isolation, with significant genetic structure and no gene flow for at least the past *ca*. 700,000 years. Their results led them to suggest that *C. oxycephalus* originated in the TEP and spread westward across the EPB to the TICP. However, they only compared populations in the TEP with those on the western edge of the EPB, not across the entire TICP. Although we also found that the mainland TEP *mt*DNA group is centrally positioned, the mitochondrial phylogenetic tree and nuclear results do not support the idea of a TEP origin. Further, the fact that the TEP group is not the most diverse genetic one in *C. oxycephalus* and that the center of diversity of the genus *Cirrhitichthys*, with seven other species, is in the Western Pacific, seems to indicate a TICP origin for *C. oxycephalus*. The EPB is a 4,000–7,000 km extension of deep water that has been present for about 65 Myr (*Grigg & Hey, 1992*; *Lessios & Robertson, 2006*). *Darwin (1872)* thought of it as an

impassable barrier due to the large deep-sea distances between islands. As with the results presented here, some recent studies found that the EPB can be crossed it remains an effective barrier that isolates populations of some organisms, including a lobster (*Panulirus penicillatus*), a benthic mollusk (*Conus ebraeus*), and a widespread Indo-Pacific pipefish (*Doryhamphus excisus*) on both sides of the EPB (*Lessios & Robertson, 2006*; *Duda & Lessios, 2009*; *Chow et al., 2011*; *Iacchei et al., 2016*). However, some transpacific reef fishes do have low levels of trans-EPB genetic differentiation (*Rosenblatt & Waples, 1986*, *Lessios & Robertson, 2006*). *Scheltema (1988)* found that species of benthic invertebrates that can cross the EPB are those likely to have an exceptionally long larval life *i.e.*, those with telepathic larvae, while other forms lacking that larval characteristic are unable to do so. The PLD has been frequently linked to the dispersal potential of marine species and hence to the degree of genetic connectivity of populations separated by long distances (*e.g.*, *Baums, Paris & Chérubin, 2006*; *Bowen et al., 2006*; *Reece et al., 2011*). *Glynn, Veron & Wellington (1996)* indicated that due to the flow rate of ocean currents across the EPB that barrier could be bridged in 60 to 120 days. *Romero-Torres et al. (2018)* using a spatially-explicit biophysical model of larval dispersal and, assuming extreme El Niño events, found the time of larval transport of 110–130 days from the Line Islands to Clipperton Atoll. The PLD of the Coral Hawkfish ranges from 36–51 days (*Brothers & Thresher, 1985*; *Robertson, Grove & McCosker, 2004*). Among members of other genera of Cirrhitids PLDs (based on very few specimens) range from ~45 to 73 days, based on very few specimens (*Brothers & Thresher, 1985*; *DeMartini, Wren & Kobayashi, 2013*). However, PLDs do vary within species, often considerably so and that of *C. oxycephalus* evidently is sufficient to have allowed it to cross the EPB in the distant past.

Most species of hawkfish (five out of seven in the IP-H-M) prefer live coral habitats, although in the Chagos Archipelago, *C. oxycephalus* apparently prefers rock-pavement habitats (*Coker et al., 2015*). However, *C. oxycephalus* is considered to be dependent on live *Pocillopora* corals on TEP reefs (*Robertson & Allen, 1996*; *Alvarez-Filip & Reyes-Bonilla, 2006*). *Cirrhitichthys oxycephalus* is the only species of this genus in the TEP and one of the few fishes strongly associated with live *Pocillopora* corals (*Robertson & Allen, 1996*; *Alvarez-Filip & Reyes-Bonilla, 2006*), which are not only common element of structural coral reefs in the TEP but also are commonly present as small, attached growths on the rocky shorelines that represent the predominant shallow reef type throughout the TEP. Reduced interspecific competition due to the absence of congeners and other coral-associated species, could have aided the establishment and spread throughout that region of *C. oxycephalus*.

## Tropical Indo-Central Pacific group

Even though many reef fishes show high levels of endemism in different TICP provinces (*Bowen et al., 2016*; *Cowman et al., 2017*), the $\Phi$ST value and the mixture of *cytb* haplotypes found in *C. oxycephalus* between provinces and populations distributed across the breadth of that vast area that extends over approximately half of the circumference of the globe, from the Red Sea, through Western Indian Ocean and the Indo-Malayan archipelago and eastern Australia, to the Line Islands and the Marquesas (*Briggs & Bowen,*

*2012*; *Toonen et al., 2016*), do not provide evidence of phylogeographic pattern across the TICP for this *mt*DNA. Although we must approach these results with caution, given the low sample sizes in some regions (*e.g.*, only four in Red Sea Province), they still offer the opportunity to formulate phylogenetic and evolutionary hypotheses.

Biogeographic studies in the distribution range of *C. oxycephalus* across TICP region have generally taken into account four semipermeable barriers, the Indo-Pacific Barrier of the Indo-Malayan archipelago, the Red Sea Barrier, the Marquesas Barrier and the mid-Indian Ocean Barrier (*Briggs, 1974*; *Gaither & Rocha, 2013*; *Borsa et al., 2016*). The first of those is considered to be a harder barrier to dispersal formed by a group of islands with large land areas that can prevent dispersal (*Craig et al., 2007*), the second is considered to be a 'softer barrier' due the unusual environmental conditions and narrowness of the Red Sea entrance (*DiBattista et al., 2016*), the third barrier is attributable to a combination of geographical isolation (a biogeographical barrier to dispersal), unusually variable sea temperatures for an equatorial archipelago (ecological distinctiveness), hydrographical isolation (cold upwelling, prevailing currents) and young geological age (few marine habitats) (*Randall, 2001*; *Gaither et al., 2010*), while the last and largest barrier to shallow-water reef fishes in the Indian Ocean is thought to be the wide expanse of deep sea west of the Lakshadweep-Maldives-Chagos archipelagoes. Many marine species that are found spread across large parts of the TICP have genetic population partitions that geographically coincide with one or more barriers between the Red Sea, the Western and mid-Indian Ocean, and the Indo-Polynesian, the Hawaiian, the Marquesas and the Easter Island Provinces (*Williams & Benzie, 1998*; *Lessios, Kessing & Pearse, 2001*; *Borsa et al., 2016*). However, the six internal barriers within the TICP seem to have had little effect on the dispersal of *C. oxycephalus* throughout that region except for the stretches of deep water isolating the Hawaiian and Easter Island Provinces from the central Pacific. Since reefs in both of those provinces harbor abundant *Pocillopora* corals (*Grigg, 1983*; *Glynn et al., 2003*; *Irving & Dawson, 2013*), the oceanic distances isolating the Hawaiian and Easter Island Provinces are less (>2,000 km) than the EPB crossed by Coral Hawkfish and given the versatility in habitat use of the Coral Hawkfish (*Coker, Pratchett & Munday, 2009*; *Coker et al., 2015*), the absence of the species in both areas is likely due to local extinction after colonization or to its failure to disperse, mediated by oceanographic conditions, across their isolating barriers, rather than ecological limitations or competition. The Coral Hawkfish joins the small group of reef fishes from a diversity of taxa known to show mismatches between genetic differentiation and other internal TICP geographic barriers, with high inter-oceanic gene flow occurring on a relatively recent evolutionary timescale across TICP ranges as large as that of *C. oxycephalus*. Those include four species of moray eels (*Reece et al., 2011*); the snapper *Lutjanus kasmira* (*Gaither et al., 2010*), four species of surgeonfishes in the genus *Naso* (*Horne et al., 2008*; *Horne & Van Herwerden, 2013*), the butterflyfish *Chaetodon meyeri* and the cornetfish *Fistularia commersonii* (*Jackson et al., 2015*; *Borsa et al., 2016*). In addition, while the snapper *Pristipomoides filamentosus* exhibits a lack of genetic structure across the sampled parts of the TICP that are also occupied by the Coral Hawkfish, it does show differentiation of the Hawaiian Island population (*Gaither et al., 2011a*). *Fistularia commersonii* is the only

reef-fish to date that has been shown to lack significant *mt*DNA structure across the entire longitudinal extent of the TIP, from the TEP to the Red Sea. However, unlike the situation in *C. oxycephalus*, in none of those 12 species has sampling included populations on both sides of all major barriers that separate all their known populations and only two of them (*Naso hexacanthus* and *F. commersonii*) have included samples from the Red Sea.

While some reef fishes with relatively short PLDs do show much more marked geographic genetic differentiation than species with longer PLDs (*Borsa et al., 2016*), PLDs among the 12 species referred to above range from relatively short (~31 days in *L. kasmira*) to very long (up to 180 days in *P. filamentosus)*. Although it has been hypothesized that the longer the PLD, the farther larvae can disperse, recent studies have concluded that this relationship does not always hold, that PLD alone is not a determining factor in successful colonization, which is likely affected by navigation, swimming ability, oceanic conditions, habitat availability in the colonized region, number of migrants and certain life history traits (*Selkoe et al., 2010*; *Leis, Siebeck & Dixson, 2011*; *Selkoe & Toonen, 2011*; *Luiz et al., 2012*; *Szabó et al., 2014*). While the lack of geographic structure across the TICP in *C. oxycephalus* could reflect ongoing gene flow facilitated by moderate-length PLD or versatility in habitat preferences that enhance dispersal within the region, it could also be due to a recent geographic expansion by a species with a large population size.

## Tropical Eastern Pacific group

Results within the TEP reveal two divergent groups with *cytb*, one composed of haplotypes largely restricted to Clipperton and the other that includes samples collected throughout the rest of the TEP islands and continent, results that are also supported by the *RAG1* $\Phi$ST test. The two TEP groups are well segregated by six mutation steps (Fig. 3) and show significant values in the $\Phi$ST for both genes, significant differentiation in *cytb* AMOVA and SAMOVA analyses and *pd* = 1.58% (Tables 1–3). Clipperton Atoll is the most isolated shoaling reef in the TEP, 950 km from the nearest emergent reef at Socorro Island in the Revillagigedos. However, the dispersal capacity of the species seems to be not strongly related to Clipperton's isolation, since a Mantel test not found correlation between genetic distances and geographic distances among the TEP samples. There are a few Clipperton haplotypes also present at the Revillagigedo Islands and continental Mexico and one from the mainland TEP *mt*DNA group at Clipperton and such apparent vagrants of other species are seen at various oceanic islands in the TEP (*Muss et al., 2001*; *Robertson, Grove & McCosker, 2004*; *Palmerín-Serrano et al., 2021*). In Hawaii, mesoscale eddy-currents transport fish eggs into entrapment centers where they are retained long enough to mature and then return to the reef of origin (*Lobel & Robinson, 1986*). Similar processes may tend to limit long-distance dispersal of *C. oxycephalus* larvae and promote the larval retention and the return of larvae to Clipperton. This process was proposed to account for the isolation of *Epinephelus clippertonensis* by *Craig et al. (2006)*, which is now known to also occur in the Revillagigedos and mainland Baja (*Robertson & Allen, 2015*), and a biophysical model by *Romero-Torres et al. (2018)* indicates potential isolation of two reef-building coral species of Clipperton based on a biophysical model of larval dispersal. Clipperton's fish fauna includes various local endemic species and oceanographic

processes that must maintain the long-term persistence of populations of those species at this island likely also produce genetically differentiated populations that may eventually become endemic species. However, other oceanographic processes in the general area of Clipperton likely enhance larval dispersal further afield, including northwards towards the Revillagigedos (*Adams & Flierl, 2010*).

**Taxonomic considerations**

*Cirrhitichthys oxycephalus* was described from type specimens collected in Indonesia. *Randall (1963)* included this species in a taxonomic review of the family and synonymized three other species with it, including *C. corallicola* (*Tee-Van, 1940*), type locality Gorgona Island, on the Pacific coast of Colombia. The original description of *C. corallicola*, which incorporated specimens from mainland sites in Mexico to Colombia but none from the TEP oceanic islands, makes no mention of *C. oxycephalus*. *Randall (1963)* synonymized *C. corallicola* with *C. oxycephalus* after examining "all available specimens of Cirrhitidae" from various museums, including the USNM. That museum houses 31 specimens collected from Gorgona that, based on the 1935 collection date, he likely assessed. Unfortunately, it is unclear from *Randall*'s *(1963)* brief reference to *C. corallicola* whether the TEP population differs morphologically from the TICP populations. However, our *cytb* results suggest that *C. corallicola* could be a valid species that encompasses all the TEP populations, as suggested by *Lessios & Robertson (2006)*, while *C. oxycephalus* extends throughout the remainder of its range across almost the entire TICP. Populations of *C. oxycephalus* in the TEP and TICP clearly need a detailed integrative systematic analysis that includes morphological and genomic assessments at both intraregional and interregional spatial scales. Such an analysis would expand the range of information about any geographic variation related to barriers within both those major biogeographic regions.

## CONCLUSIONS

This work revealed a large-scale phylogeographic pattern in *C. oxycephalus* across the Indo-Pacific range of the species based on mitochondrial *cytb*, which was partially supported by the *RAG1* nuclear marker. That pattern, which includes three largely allopatric genetic groups, is related primarily to the effects of the East Pacific Barrier, the world's widest (4,000+km) open ocean barrier to migration of tropical reef fishes. Our results agree with previous studies that indicated strong genetic differences between populations of some widely distributed reef fishes between the TICP and TEP and support the potential resurrection of *C. corallicola*, type locality Colombia, which was synonymized with *C. oxycephalus* (type locality Indonesia) by *Randall (1963)*. Within the TICP, however, known barriers between the Indian Ocean and west Pacific, and large distances separating isolated islands and archipelagos in those two oceans evidently have had no equivalent effects in phylogeographic differentiation among *C. oxycephalus* populations, although such may be detectable using more sensitive genetic markers. In contrast, we did find phylogeographic differentiation within the TEP, between Clipperton and the rest of that region. Clipperton, which has a number of endemic reef fishes, is the most isolated reef

in the entire TEP, which probably accounts for the isolation of its Coral Hawkfish population. This Clipperton-specific result adds to the range of patterns of genetic differentiation of conspecific populations found in the offshore islands and mainland of the TEP.

## ACKNOWLEDGEMENTS

We thank Diego Inclan, Johanna Moreira, Esteban Herrera, Issac Chinchilla, Carmen Pedraza, Georgina Palacios, Salvador Romero, Paola Torres, Omar Valencia, Cecilia Mesen, Francisco Martínez, Paola Palmerín, Beatriz Naranjo, Xavier Madrigal, Yareli Lopéz, Edgar Acevedo and Kang-Ning Shen, for their collaboration in the collection process. Maried Ochoa for the assistance on analysis on RStudio and Oscar J. Ruiz Maraver for help with QGis map. Thanks to CIPRES Cyberinfrastructure for Phylogenetic Research XSEDE for their computational support. Thanks to the Mexican Navy that provided generous logistical support to work on Revillagigedo Archipelago (Socorro and Clarion islands).

### Funding

Rolando Quetzalcoatl Torres-García was supported by a MsC scholarship from the Consejo Nacional de Ciencia y Tecnología (CONACyT). This work was supported by funding provided for CONACyT (grant No. CB-2014-240875) and the Universidad Michoacana de San Nicolás de Hidalgo (CIC-UMSNH 2013-2020). The funders had no role in study design, data collection and analysis, decision to publish, or preparation of the manuscript.

### Grant Disclosures

The following grant information was disclosed by the authors:
Consejo Nacional de Ciencia y Tecnología: CONACyT.
CONACyT: CB-2014-240875.
Universidad Michoacana de San Nicolás de Hidalgo: CIC-UMSNH 2013-2020.

### Competing Interests

The authors declare that they have no competing interests.

### Author Contributions

- Rolando Quetzalcoatl Torres-García conceived and designed the experiments, performed the experiments, analyzed the data, prepared figures and/or tables, authored or reviewed drafts of the article, and approved the final draft.
- Michelle R. Gaither conceived and designed the experiments, performed the experiments, analyzed the data, prepared figures and/or tables, authored or reviewed drafts of the article, provide samples and collection permits, and approved the final draft.

- D. Ross Robertson conceived and designed the experiments, performed the experiments, analyzed the data, prepared figures and/or tables, authored or reviewed drafts of the article, and approved the final draft.
- Eloisa Torres-Hernández conceived and designed the experiments, performed the experiments, analyzed the data, prepared figures and/or tables, authored or reviewed drafts of the article, and approved the final draft.
- Jennifer E. Caselle conceived and designed the experiments, prepared figures and/or tables, authored or reviewed drafts of the article, and approved the final draft.
- Jean-Dominique Durand conceived and designed the experiments, prepared figures and/or tables, authored or reviewed drafts of the article, and approved the final draft.
- Arturo Angulo conceived and designed the experiments, prepared figures and/or tables, authored or reviewed drafts of the article, provide samples and collection permits, and approved the final draft.
- Eduardo Espinoza-Herrera conceived and designed the experiments, prepared figures and/or tables, authored or reviewed drafts of the article, provide samples and collection permits, and approved the final draft.
- Francisco J. García-De León conceived and designed the experiments, prepared figures and/or tables, authored or reviewed drafts of the article, and approved the final draft.
- Jonathan Valdiviezo-Rivera conceived and designed the experiments, prepared figures and/or tables, authored or reviewed drafts of the article, provide samples and collection permits, and approved the final draft.
- Omar Domínguez-Domínguez conceived and designed the experiments, performed the experiments, analyzed the data, prepared figures and/or tables, authored or reviewed drafts of the article, and approved the final draft.

**Animal Ethics**

The following information was supplied relating to ethical approvals (*i.e.*, approving body and any reference numbers):

Organism collection was supported and allowed by the following institutions and permits by Mexican Ministry of Environment and Natural Resources under collection permits numbers (PPF/DGOPA-035/15; and F00.DRPBCPN.DIR.RBAR-100/2015-CONANP), and for El Salvador (MARN-AIMA-004-2013), Panama (SC/A-17-19), Costa Rica (007-2013-SINAC and R-056-2105-OT-CONAGEBIO), and Ecuador (013/2012 PNG; N°21-2017-EXP-CM-2016-DNB/MA and MAAE-DBI-CM-2021-0152).

**Field Study Permissions**

The following information was supplied relating to field study approvals (*i.e.*, approving body and any reference numbers):

Organism collection was supported and allowed by the following institutions and permits by Mexico (PPF/DGOPA-035/15; and F00.DRPBCPN.DIR.RBAR-100/2015-CONANP), El Salvador (MARN-AIMA-004-2013), Panama (SC/A-17-19), Costa Rica (007-2013-SINAC and R-056-2105-OT-CONAGEBIO), and Ecuador (013/2012 PNG; N°21-2017-EXP-CM-2016-DNB/MA and MAAE-DBI-CM-2021-0152).

## DNA Deposition

The following information was supplied regarding the deposition of DNA sequences:

The cytb sequences are available at GenBank: PP328592 to PP328757.

The accession code for RAG1 are available at Zenodo: Torres-Hernández, E., Domínguez´Domínguez, O., Torres-García, R. Q., Gaither, M. R., Robertson, D. R., Caselle, J., Durand, J.-D., Angulo, A., Espinoza-Herrera, E., García-De León, F. J., & Valdiviezo-Rivera, J. (2025). Geographic genetic variation in the Coral Hawkfish, *Cirrhitichthys oxycephalus* (Cirrhitidae), in relation to biogeographic barriers across the Tropical Indo-Pacific. In Geographic genetic variation in the Coral Hawkfish, *Cirrhitichthys oxycephalus* (Cirrhitidae), in relation to biogeographic barriers across the Tropical Indo-Pacific. Zenodo. https://doi.org/10.5281/zenodo.11323581. Both alignments are available in the Supplemental File.

## Data Availability

The DNA sequences for the gene mitochondrial gene *cytb* and nuclear *RAG1* are available in the Supplemental Files.

## Supplemental Information

Supplemental information for this article can be found online at http://dx.doi.org/10.7717/peerj.18058#supplemental-information.

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
