# Peer review of "Geographic genetic variation in the Coral Hawkfish, Cirrhitichthys oxycephalus (Cirrhitidae), in relation to biogeographic barriers across the Tropical Indo-Pacific"

_PeerJ, doi:10.7717/peerj.18058_

## Round 0.1 · original submission · Major Revisions

Dear Authors,

We have completed the review process of your manuscript titled "Geographic genetic variation in the Coral Hawkfish, Cirrhitichthys oxycephalus (Cirrhitidae), in relation to biogeographic barriers across the Tropical Indo-Pacific." After careful evaluation by three reviewers, we have decided that your manuscript requires major revisions to be considered for acceptance.

Reviewer 1 noted that your paper provides valuable insights into the population genetics and biogeography of the Coral Hawkfish. They appreciated the extensive sampling effort and the clarity of your results. However, they suggested that incorporating more molecular markers could enhance the robustness of your findings. Reviewer 2 also commended the thoroughness of your study but raised concerns about certain methodological aspects, such as the use of RAG1 data and the interpretation of results. They recommended addressing these issues in your revision. Reviewer 3 highlighted several strengths of your manuscript, including its contribution to our understanding of reef fish phylogeography. However, they also identified areas for improvement, such as the clarity of writing, the definition and use of acronyms, and the discussion of demographic history and ecological factors.

In summary, the reviewers found your study to be well-conducted and potentially valuable to the field. However, they identified several areas where clarification and additional analyses are needed to strengthen the manuscript. We encourage you to carefully address each of the reviewers' comments and resubmit your revised manuscript.

Thank you for considering our journal for the publication of your research.

Sincerely,

Armando Sunny

Reviewer 1 ·

Basic reporting

This is a very 'classic' paper of population genetics / biogeography of a tropical reef fish.

Sampling is very extensive and covers the entire range fo the species, in that respect, it would be very difficult to do better.
Molecular markers are minimal, in this genomics era, as they are based on few mt and n loci.
That said, results are very clear and shed some important light on the system. This is one of the cases where the results are important and relevant, thus making the actual approach less important, as long as it is cogent and well analyzed (which it is)

Experimental design

Sampling is very extensive, with large sample sizes and a very wide geographic range that encompasses the entire distribution of the species.
Molecular markers are minimal, but they cover both mitochondrial and nuclear loci.

Validity of the findings

I find the results very convincing. They provide an interesting hypothesis to be tested in the future by anyone who wants to venture in the genomics realm. I very much doubt major differences will be uncovered, but some important details and modes of evolution will likely be found.

Additional comments

Overall the paper is very well written and could be accepted as is

Reviewer 2 ·

Basic reporting

The article is very well written, well organized, and overall it flows very well. The literature cited is thorough, and the objectives are clear.

Experimental design

The experimental design overall is fine. The analyses are appropriate for the study. The decision to include the RAG1 data is bit strange, as it is only a very small fragment of a very slow evolving gene, and only present for about 1/3 of the samples in the study, but otherwise all is fine.

Validity of the findings

While based only a short segment of mitochondrial DNA, the findings corroborate those of prevouis studies using different loci (albeit still mtDNA). They are interpreted in a way to makes logical sense, and they tell a compelling story about the extent to which this species is able to, or not able to, pass semipermeable barriers to dispersal in the ocean.

Additional comments

Overall the authors have conducted a scientifically a solid study that contributes to our growing knowledge of reef fish phylogeography and speciation -- especially in the Tropical Eastern Pacific. In the future I would be curious to see if the authors use more cutting edge genomics tools, whether there will be additional genetic breaks in the Indo-Pacific, especially with the Red Sea and the Sunda Shelf barriers.

Line 67 remove comma after availability
Line 232 – You cannot have 165 out of 163. I think you must have these numbers reversed.
Line 198 and Figure 2 – the tree is not actually rooted with the outgroup. It is shown unrooted. If you are going to include the outgroup, you should use that branch to root the tree.

Just as an aside, I have never seen the abbreviation Dp used for pairwise genetic distance. I had to go back to find out what that meant. All of the other acronyms and abbreviations used here are pretty common convention in population genetics, but Dp was new to me.

Table S2 – what does the bold text indicate?

Table S1 – Why are there no GenBank numbers for the Rag1 sequences? If they are not deposited there, you should delete that column in the table as it is completely blank.

Lines 265-269 – I’m not sure what you mean with these sentences. Please add detail or clarify. You say “The three-group arrangement of Briggs & Bowen had the highest percentage within populations 89.19% (Table 1).” 89.19% of what? That number does not appear in Table 1.

Lines 317-329. I would also chalk up the differences to the fact that so few bp of Rag1 were analyzed. I wouldn’t expect to see differences among populations with so few sites analyzed.

Throughout – I’ve not heard of mitochondrial genes referred to as “mitogenes”. Is this a commonly used term? I cannot find any phylogenetic or population genetic papers that use this.

Discussion – I would love to see some mention of Hawaii and Pitcairn-Easter Island in the discussion. There is obviously dispersal potential to reach those islands, but I’d be curious to hear about why the authors believe the species is not there. They certainly seem to be able to deal with lots of different conditions – reefs in the TEP are nothing like those in the TICP, yet the hawkfish is common in both areas. With regards to Hawaii and Pitcairn/Easter, small, isolated areas with strong barriers to gene flow may be more susceptible to local extinction, and maybe it was there at some point and is no longer? It’s obviously all just speculation but those island chains are a piece of the overall picture that are not mentioned at all in the discussion.

Reviewer 3 ·

Basic reporting

This manuscript examines population genetic structure in a widespread coral reef fish using mitochondrial and nuclear DNA. In regards to basic reporting, the manuscript is mostly well written. However, there are certain sections that are slightly cumbersome to read or difficult to interpret. My main complaint about the writing, though, is the use of acronyms. There are several acronyms that I do not believe were ever defined (RSP, IPHMEIP, WIOP) and others that were defined but never used again after definition (ex LMC). While a reader can go through and deduce what each acronym means, I suggest the authors do a careful review of their use of acronyms and make sure all are necessary, and all that are necessary are defined.

Regarding literature references and background I believe that the authors sufficiently summarized relevant studies for the reader to easily follow the study. All figures and tables are helpful and none seem redundant.

Experimental design

In terms of experimental design, there are some areas that one can find shortfalls. However, I don’t believe that these negate the study, but I do believe that these caveats could be discussed in the manuscript more than they are. One of these shortfalls is sample size, which I’m of two minds with. Figure 1 clearly shows sample sizes at each locality. On one hand, the geographic scope of sampling is very impressive. It is no easy feat to get samples from all of those localities. On the other hand, the numbers per locality are very low for most (ex. 1 in Australia, 3 in Reunion, etc). Typically for Sanger phylogeography I like to see ~30 samples per site for marine fishes with very large population sizes so that you can detect rare alleles. This falls far below that. Does that mean that the results are not publishable? In my opinion, no. However, I do think that they need to discuss that caveat more. Additional samples that detect rare alleles could help show more structure across the Indo-West Pacific. I don’t think they need to go back and sample more because field work is very expensive and unpredictable, but I do think it needs to be mentioned. This is also relevant to the Rag1 dataset, which is even fewer samples and additionally much fewer base pairs as the section they amplified is quite short (which really needs to be mentioned).

Importantly, the analyses conducted and their interpretations are appropriate. But as with all studies, there are some shortfalls that should at least be mentioned.

Validity of the findings

The findings of this study do build upon previous work of one of the coauthors. However, it is barely even discussed how the two studies relate - only three sentences in the discussion. Are phiST values similar between studies? Do they complement one another? Do they disagree in any way? The authors clearly find strong structure between the eastern Pacific and the Indo-West Pacific. However, I feel like there could be improvement in how they explain this finding. They spend a lot of space talking about pelagic larval duration, deep water gaps, and dispersal. However, these things clearly do not dictate the structure in this species because you can see different scenarios around deep water/large habitat gaps: 1) there are deep water gaps in the Indian Ocean that result in zero structure, 2) deep water gaps in the eastern Pacific that result in significant structure, and 3) deep water gaps in the Pacific to Hawaii that result in zero establishment. So it’s clearly more than just habitat gaps as you can see the entire gamut of outcomes surrounding large habitat gaps. The authors do mention briefly the importance of Pocillopora coral, but there are other ecological factors that likely play a role in the distribution and structure of this species as well, such as competition. And while the authors comment a lot on PLD and distance, there isn’t much discussion of prevailing currents across many of the barriers. Perhaps the most important omission in the discussion, though, is of demographic history. Given the haplotype and nucleotide diversity estimates the authors present, it is highly suggestive of a large population bottleneck followed by population expansion, which has been found in a variety of marine organisms. This has likely had a profound impact on your results and can explain a lot of what you find and I think should be discussion in far more detail.

Additional comments

Overall I like this study for several reasons. First, it’s a very rare case where we get scientific replication in phylogeographic studies, and replication is a central pillar of science. Secondly, while it uses the same focal species and similar methods to a previous study, it builds upon that earlier study in terms of geographic scope, opening new questions. Third, it is straight forward, the analyses are appropriate, and the discussion for the most part is in-line with the data. However, the discussion could be improved and is missing certain critical pieces. Broadening the geographic scope also limited the availability of sufficient sampling. This, and other caveats, should be mentioned and discussed in the discussion more than they are currently. Secondly, the discussion doesn’t fully explain the results obtained, and I think important points about demographic history, competition, and currents are missing. By making improvements on the Discussion I believe the entire manuscript will become much better and make an important contribution to the literature.

---

## Round 0.2 · accepted · Accept

Dear Authors,

I am delighted to inform you that both reviewers have confirmed that your revisions were satisfactory. Your manuscript has been accepted for publication in PeerJ.

Congratulations!

Best regards,
Armando Sunny

Reviewer 2 ·

Basic reporting

As was the case with my first review of the manuscript, I found the article to be well written with clear objectives.

Experimental design

Same as the first review. The design is appropriate for the objectives, and the breadth of the sampling is impressive.

Validity of the findings

The findings seem valid and represent a significant contribution towards our understanding of the biogeography of fishes in the region.

Additional comments

The authors have addressed all of my concerns from the original review, which were relatively minor at that point. I am very happy to see the discussion added regarding the absence of the species in Hawaii and Easter Island. I look forward to seeing the future work from this group on the genome-wide patterns of genetic variation in hawkfishes, as well as (hopefully) some integrative taxonomic investigation of the TEP lineages to determine whether they differ morphologically.

Reviewer 3 ·

Basic reporting

The basic reporting, including clarity, language used, references and structure is easy to follow and interpret. I see no issues in basic reporting other than a few minor edits that can be made during article proofs (one example: on page 16 of the PDF line 375 you repeat "based on very few specimens" twice in one sentence). This version is improved from previous versions by removing unnecessary and undefined acronyms.

Experimental design

The analyses and design of the study are well thought out and the methods appropriate. The sampling remains impressive, but I am glad that the authors did mention the lower sample sizes in the Results section.

Validity of the findings

I respect the authors' interpretation of their results and largely agree with them. The Discussion is clear and the conclusions are easy to follow. I look forward to seeing any follow up studies on the taxonomy of the species.

Additional comments

Overall I am happy with the changes that the authors have made to the manuscript. I believe that this manuscript will make a nice addition to the literature and hope to see it cited and widely read in the future.